# A systems biology approach unveils different gene expression control mechanisms governing the immune response genetic program in peripheral blood mononuclear cells exposed to SARS-CoV-2

**Damariz Marin**[1], **Geysson Javier Fernandez**[2], **Juan C. Hernandez**[3,4], **Natalia Taborda**[5]*

1 GIOM, Facultad de Odontología, Universidad Cooperativa de Colombia, Medellín, Colombia, 2 Biología y Control de Enfermedades Infecciosas (BCEI), Universidad de Antioquia- UdeA, Medellín, Colombia, 3 Infettare, Facultad de Medicina, Universidad Cooperativa de Colombia, Medellín, Colombia, 4 Grupo Inmunovirología, Facultad de Medicina, Universidad de Antioquia- UdeA, Medellín, Colombia, 5 Corporación Universitaria Remington, Programa de Medicina, Facultad de Ciencias de la Salud, Corporación Universitaria Remington, Medellín, Colombia

* nataliataborda@gmail.com

## Abstract

COVID-19 and other pandemic viruses continue being important for public health and the global economy. Therefore, it is essential to explore the pathogenesis of COVID-19 more deeply, particularly its association with inflammatory and antiviral processes. In this study, we used the RNA-seq technique to analyze mRNA and non-coding RNA profiles of human peripheral blood mononuclear cells (PBMCs) from healthy individuals after SARS-CoV-2 *in vitro* exposure, to identify pathways related to immune response and the regulatory post-transcriptional mechanisms triggered that can serve as possible complementary therapeutic targets. Our analyses show that SARS-CoV-2 induced a significant regulation in the expression of 790 genes in PBMCs, of which 733 correspond to mRNAs and 57 to non-coding RNAs (lncRNAs). The immune response, antiviral response, signaling, cell proliferation and metabolism are the main biological processes involved. Among these, the inflammatory response groups the majority of regulated genes with an increase in the expression of chemokines involved in the recruitment of monocytes, neutrophils and T-cells. Additionally, it was observed that exposure to SARS-CoV-2 induces the expression of genes related to the IL-27 pathway but not of IFN-I or IFN-III, indicating the induction of ISGs through this pathway rather than the IFN genes. Moreover, several lncRNA and RNA binding proteins that can act in the cis-regulation of genes of the IL-27 pathway were identified. Our results indicate that SARS-CoV-2 can regulate the expression of multiple genes in PBMCs, mainly related to the inflammatory and antiviral response. Among these, lncRNAs establish an important mechanism in regulating the immune response to the virus. They could contribute to developing severe forms of COVID-19, constituting a possible therapeutic target.

**Data Availability Statement:** The data underlying the results presented in the study are available from: 10.6084/m9.figshare.26790304.

**Funding:** This work was supported by Universidad Cooperativa de Colombia, Universidad de Antioquia and Corporación Universitaria Remington. The funders had no role in study design, data collection and analysis, decision to publish, or preparation of the manuscript.

**Competing interests:** The authors have declared that no competing interests exist.

# Introduction

COVID-19, alongside other pandemic viruses, continues being important for both public health and the global economy. This complex respiratory disease presents a myriad of potentially life-threatening complications, including acute respiratory distress syndrome (ARDS), pneumonia, disseminated intravascular coagulation, acute cardiac injury, and multiorgan failure [1, 2]. Compounding the challenge, the emergence of novel variants such as Omicron has heightened concerns due to their increased transmissibility and a rise in hospitalizations and intensive care unit admissions, even in countries with high vaccination rates [3, 4]. This indicates an ongoing evolution of SARS-CoV-2 and the disease it causes, underscoring the importance of epidemiological surveillance and the development of adjunctive treatments.

In light of these challenges, it is important to consider the pathogenesis of COVID-19, particularly its association with and exacerbate inflammation [5, 6] and altered antiviral response [7]. Specifically, dysregulated systemic inflammatory responses, known as cytokine storms, have been closely linked to the severity of infection [8, 9]. Cytokine storm is a major contributor to ARDS and multiorgan failure, significantly impacting patient outcomes [9, 10]. Related with this, it has been described that the interaction between epithelial cells and peripheral blood mononuclear cells (PBMCs) during *in vitro* SARS-CoV-2 infection is significant in producing pro-inflammatory cytokines such as IL-6 and IL-1β [11], 2023 #8}. Therefore, elucidating the cellular and molecular mechanisms involved in the response to SARS-CoV-2 is essential for understanding its pathogenesis and potential therapeutic targets in COVID-19.

On the other hand, a low production of IFN type I and III has been described in severe patients with COVID-19 compared to patients with mild symptoms or other infectious diseases [12]. Associated with this, different studies have identified accessory proteins of SARS-CoV-2 that can inhibit the production and signaling of type I IFNs, such as Orf9b, PLpro (the papain-like protease), Nsp1 and Orf3b [13–16]. However, there are contradictory studies where an exacerbated production of IFN-I has been found in severe patients with COVID-19, which is why it has been proposed that it is the imbalance in the production of these molecules during the different stages of the disease, which can lead to the development of severe symptoms and unfavorable outcomes [17]. With a delayed production in the initial phases of the infection, which favors viral replication; and subsequently, this high viral load induces a high production of IFN-I in monocytes, macrophages and dendritic cells, which not only leads to the production of ISGs, but also contributes to the exacerbated production of inflammatory mediators, having deleterious effects [17]. Additionally, it has been reported that the IL-27 pathway can induce an antiviral response during various viral infections, including SARS-CoV-2; by activating the STAT1 pathway and producing different ISGs independently of IFN-I [18–20]. However, the heterogeneity between patients and the levels of the different inflammatory and antiviral markers suggests that other mechanisms are involved in this imbalance and the development of severe forms of COVID-19.

Therefore, understanding the transcriptomic foundation of the SARS-CoV2 and PBMCs interaction process is decisive, as PBMCs play a pivotal role in responding to and clearing viral infections like SARS-CoV-2 [21]. In line with this, the SARS-CoV-2 infection in PBMCs has been previously studied, revealing insights into the immune mechanisms involved. Studies have identified aberrantly expressed mRNA and lncRNA biomarkers associated with SARS-CoV-2 severity, impacting disease outcomes [22–24]. Despite PBMCs not being susceptible to direct SARS-CoV-2 infection, exposure to the virus triggers a JAK/STAT-dependent innate immune response, particularly in monocytes, leading to transcriptional changes indicative of a global innate immunity-related gene profile [25]. However, while prior research has explored the transcriptomic profile of PBMCs in COVID-19 patients, providing a better understanding

of the immune response, these studies often focused solely on describing the changes on levels of the mRNAs. Nevertheless, our understanding of the gene expression patterns and molecular mechanisms regulating the response of PBMCs to SARS-CoV-2 exposure remains limited. These mechanisms encompass factors like DNA-binding proteins and long non-coding RNAs (lncRNAs) responsible for mRNA synthesis, while others like RNA-binding proteins (RBPs) govern mRNA degradation.

Taking this into account, this study investigates the mRNA and non-coding RNAs profiles PBMCs following exposure to SARS-CoV-2. This analysis identified a distinct set of differentially expressed genes associated with various biological processes, including immune response, signaling pathways such as IL-27, cell differentiation and metabolism. Additionally, this research sheds light on the regulatory mechanisms governing gene expression by comprehensively examining both transcriptional and post-transcriptional control of genes involved in IL-27 pathway. This involves elucidating the roles of transcription factors, lncRNAs, and RBPs orchestrating gene expression dynamics. This deeper understanding could be key for developing novel therapeutic strategies targeting specific stages of the immune response during COVID-19.

## Materials and methods

### PBMCs and SARS-CoV-2 viral stock

Peripheral blood mononuclear cells were obtained from six healthy male volunteers, between 20 and 40 years old were included, following the previously described protocol [26]. Individuals who reported any illness in the previous 4 weeks, chronic diseases, long-term medication use, cancer, illicit drug use or mental disorders were excluded. The recruitment period was from 14[th]/May/2023 to 6[th]/July/2023. The viral stocks (D614G, EPI_ISL_536399) were obtained in the VERO E6 cells and quantified by the plaque formation method [27].

For stimulation, $3x10^6$ PBMCs were seeded in RPMI-1640 supplemented with FBS (5%), penicillin-streptomycin (1%) and L-glutamine (2mM). The cells were exposed to the virus at a MOI (multiplicity of infection) of 0.1 and incubated for 24 hours at 37˚C and 5% $CO_2$. Cells without exposure to the virus were used as controls.

### Total RNA extraction, library construction, and RNA-seq analysis

Total RNA extraction was conducted using the Direct-zol RNA MiniPrep kit (Zymo Research) following the manufacturer's protocol as previously described [26]. RNA integrity was assessed via agarose gel electrophoresis to ensure that the 28S to 18S rRNA ratio exceeded 1.8. Additionally, all samples underwent analysis on the tape station to quantify an RNA integrity number greater than 7. For mRNA sequencing library preparation, six samples from the PBMC following exposure to SARS-CoV-2 and six from the uninfected PBMCs group were utilized, with TruSeq Stranded mRNA Sample Prep Kits (Illumina, USA) and 600 ng of total RNA. These samples were indexed with adaptors and sequenced using a HiSeq 4000 instrument (Illumina, USA) for single-end sequencing. RNA-Seq experiments were sequenced with a depth ranging from 20 M to 30 M reads per sample.

### Read alignment and differential gene expression analysis

The FASTQ files containing sequence data were processed as mentioned before [28]. Briefly, data were preprocessed with Trimmomatic and Prinseq functions, setting an average Phred score >20. Subsequently, alignment with the reference genome (Homosapiens GRch38.14) was performed with the TopHat2 function and counts for RefSeq genes were obtained with

HTseq. The content matrix was normalized using the DESeq2 v.3.20 package in R v.4.2.2. The mRNAseq mapping results obtained expression profiles of both mRNAs from the RefSeq databases and lncRNAs from NONCODE. mRNAs and lncRNAs, were considered differentially expressed if |fold change| (FC) $\geq$ 2 and FDR $\leq$ 0.05. Transcripts per million (TPM) were used to assess mRNA and lncRNA abundance.

### Transcription factor motif analysis

In this study, we employed several bioinformatics tools to investigate the transcription factor motifs associated with differentially expressed genes. Initially, we retrieved a comprehensive list of transcription factors in the human genome from the Human Protein Atlas database [29]. Subsequently, we utilized EnrichR (ChEA database) [30] and iRegulon [31] tools to perform transcription factor enrichment analysis.

### Gene enrichment analysis

Enrichr was employed for gene enrichment analysis using the Biological Process (Gene Ontology; GO) databases. The enrichment results were represented as a percentage of genes per term computed by Enrichr as overlap. Interferome v2.0 was utilized for the identification of interferome genes [32], while the RNA-binding protein specificities (RBPDB) database provided information on RNA-binding proteins [33].

### Ethics

All individuals read and signed a written consent report, previously reviewed and approved by the ethics committee from Universidad Cooperativa de Colombia (Act 001–2022). The experiments were performed according to the principles of the Declaration of Helsinki.

## Results

### SARS-CoV-2 induces a different transcriptional profile in PBMCs

PBMCs were exposed to SARS-CoV-2 for 1 h, 24 h, 48 h, and 72 h; no changes in viral RNA were found between time points indicating non-productive viral replication in these cells (Fig 1a). To get insight if the PBMCs exposed to the SARS-CoV-2 virus could induce changes in gene expression we conducted a transcriptome analysis, revealing 14210 transcripts. After filtering by log2 fold change > 1.0 and FDR (<0.05) was obtained 790 differentially expressed genes (DEGs) of which 733 correspond to mRNAs and 57 to non-coding RNAs. A principal component analysis (PCA) was performed to reduce dimensionality and visualize whether exposure to SARS-CoV-2 induces a different transcriptional profile compared to mock-infected cells. It is observed that there is a distinct genetic profile that differentiates control PBMCs from SARS-CoV-2-exposed PBMCs (Fig 1b). Furthermore, according to Fig 1b and 1c, 555 positively regulated genes and 178 negatively regulated genes were identified among the DEGs. It can be observed that the expression of the positively regulated genes in individual 1 has a greater intensity compared to the other individuals (Fig 1d). Even more, it was observed among the genes with the highest coefficient of variation (greater than 180%) that they are mainly associated with inflammatory and antiviral responses (Fig 1e).

### Exposure to SARS-CoV-2 induces a mainly inflammatory response

Next, we conducted an enrichment analysis to investigate the cellular processes associated with the differentially expressed genes (DEGs) in response to the SARS-CoV-2 stimulus, finding 19 terms grouped into 5 main processes defined as an immune response, acute phase

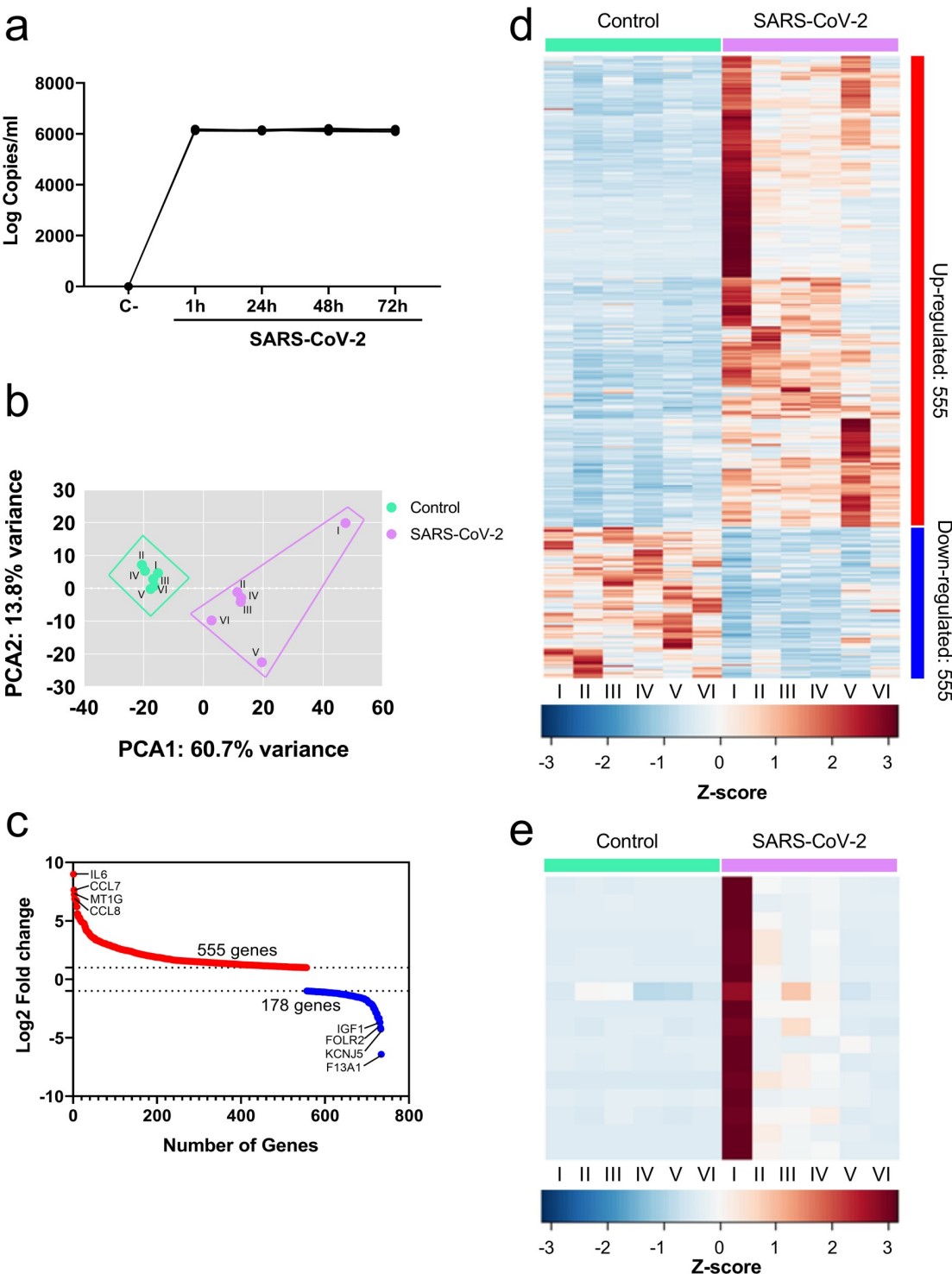

**Fig 1. SARS-CoV-2 induces a different transcriptional profile in PBMCs. (a)** Real-time PCR for viral protein E from supernatant's PBMCs exposed to SARS-CoV-2. **(b)** Principal component analysis of gene expression from PBMCs exposed to SARS-CoV-2. The percentage of the variance of each principal component (PC1 and PC2) for control (I—VI) and SARS-CoV-2 exposed cells (I—VI). (c) the number of genes vs log2 fold change. (d) Heatmap of 734 Z-score normalized differentially expressed genes (DEG) of control and SARS-CoV-2-exposed cells. (e) Heatmap of 16 Z-score normalized differentially expressed genes with the highest coefficient of variation of control and PM-exposed cells.

response, signaling, cell proliferation and metabolism (Fig 2a). Within these, the immune response groups the largest number of terms such as inflammatory response, cellular response to interferon-gamma, signaling pathways related to cytokines, neutrophil-mediated immunity, and regulation of IL-1b production Furthermore, genes with the highest regulation within each term were highlighted, including the positive regulation of genes associated with the immune response, such as CCL8, GBPI, ICAM1, IL6, THBS1, IRAK2 and LLRB2. In addition, metabolism-related genes, including IDO1, IDO2, HK2 and HIF4, were regulated. As well as the regulation of ACOD1, PARPG, SOCS3, STAT1 and OAS3, related to cell signaling processes (S1 Fig).

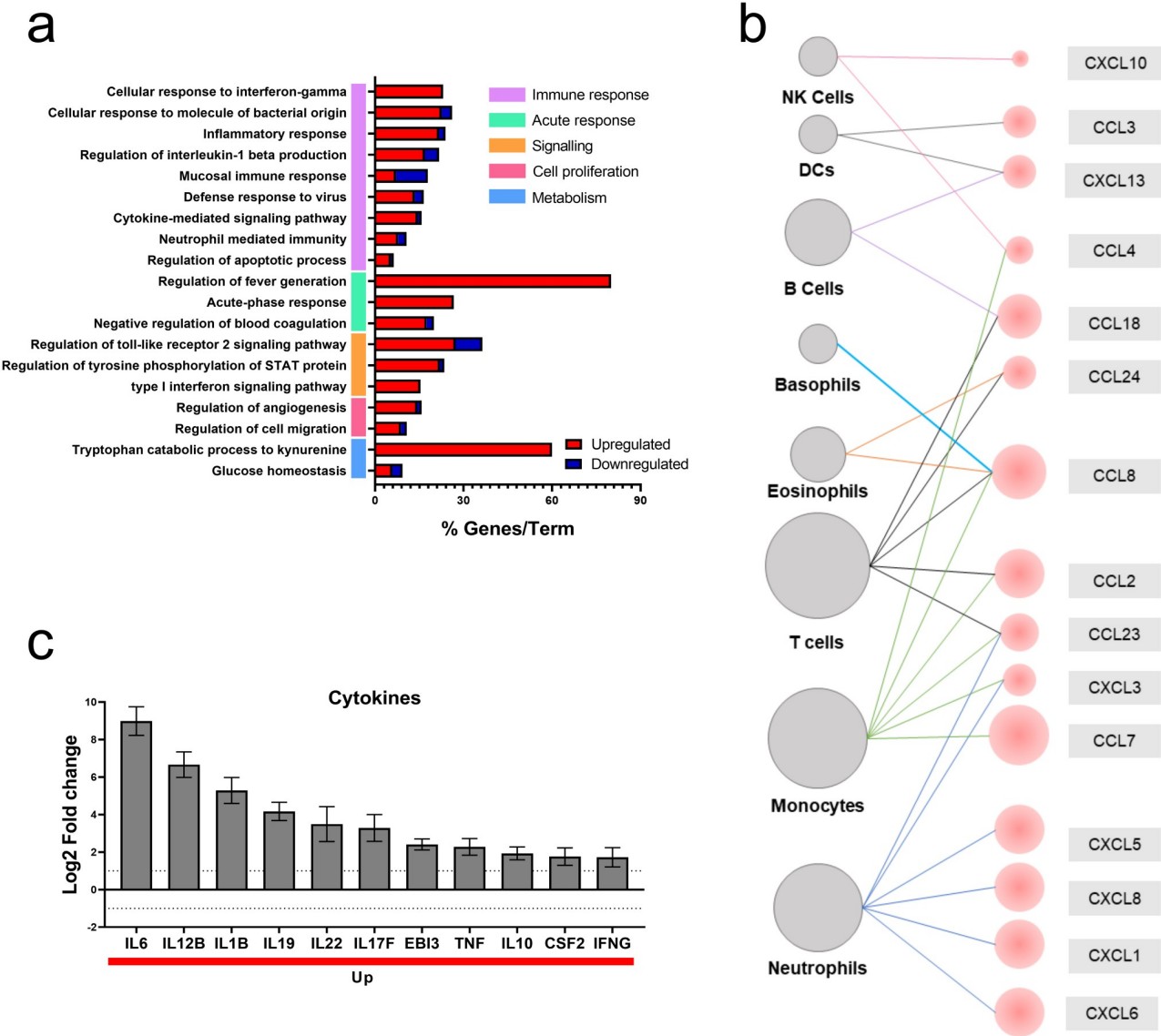

**Fig 2. SARS-CoV-2 stimulus induces an inflammatory response in PBMCs.** (a) Gene-term enrichment analysis of differentially expressed genes (DEG) in PBMCs exposed to SARS-CoV-2. The red and blue colors of the horizontal bars indicate the up- and down-regulated genes, respectively. And their length indicates the percentage of regulated genes within the term (% genes/term). The main groups of gene terms are indicated by the colored vertical bars. (b) The cells recruited in response to SARS-CoV_2 exposure. Red color indicates up-regulated genes and dot size indicates the magnitude of log2 -fold change. (c) Bar plot of cytokines regulated during SARS-CoV-2 exposure. Red bars indicate up-regulated genes.

Additionally, the positive regulation of multiple chemokines was observed, such as CCL7, CCL8, CXCL5, CXCL8, CXCL5, CCL18, CCL2, among others. Cells such as T cells, monocytes and neutrophils are the main targets (Fig 2b). On the other hand, multiple positively regulated cytokines were observed, with IL-6 being the one with the highest fold change, followed by IL-12B, IL1B, IL19, IL22, among others (Fig 2c).

## Exposure to SARS-CoV-2 in PBMCs induces the expression of interferon genes

Considering that an altered antiviral response has been described during SARS-CoV-2 infection, the expression of interferon genes and interferon response genes (ISGs) was explored in PBMCs exposed to the virus. 96 ISGs were observed, corresponding to 89 positively regulated and 7 negatively regulated genes (Fig 3a). Specifically, ISGs such as BATF2, GBP1, IFIT3, IRF7, LAMP3, OASL and RSDA2, among others, were found (Fig 3b). Despite observing the regulation of multiple ISGs, only 3 significantly regulated interferon genes were found, EBI3, IFNG and IL27 (Fig 3c). The fold change of both the interferon genes and the ISGs individually is presented in S2 Fig, where it can be observed that individual 1 has a differential expression pattern concerning the other individuals (S2 Fig).

## Transcription factors and RNA binding proteins during SARS CoV-2 exposure

Additionally, we explored whether exposure of PBMCs to SARS-CoV-2 could affect the expression of transcription factors (TFs). The regulation of 48 TFs was found, of which 43 had positive regulation and 5 had negative regulation (Fig 4a). Among these, factors are ATF3, BCL6, FOSB, FOSL1, IRF7, JUN, RELB, and STAT1, with the inflammatory response being the main process to which they are associated. Furthermore, the transcription factors involved in the pathways that regulate the expression of the DEGs obtained were investigated using the Enrich R tool. It was identified that TFs such as RELB, IRF8, JUN, FOSL1, RUNX1, IRF1, and STAT1 are associated with the main biological processes that group the largest number of genes, among which are the inflammatory response, antiviral response, metabolism and signaling (Fig 4b).

Moreover, it was explored whether exposure to SARS-CoV-2 affects the expression of RNA binding proteins (RBP), which regulate the expression of other genes. In total, 4 positively regulated RBPs were found: PARP12, ZC3H12C, RBM47 and ZC3H12A (Fig 5a and 5b). These proteins are involved in functions such as ADP-ribosylase, endoribonuclease, splicing, stabilization and editing of RNA (Fig 5c).

## SARS-CoV-2 regulates the expression of long non-coding RNAs, pseudogenes, antisense RNA and miRNAs

Finally, we explored whether exposure to SARS-CoV-2 can modulate the expression of non-coding RNAs, as a possible mechanism of gene regulation. Fifty-seven differentially expressed non-coding RNAs were found, which are distributed in the categories of long non-coding RNAs (lncRNA; 52.6%), pseudogenes (21.1%), antisense RNA (21.1%) and micro-RNAs (miRNA; 5.2%) (Fig 6a). Furthermore, a characterization was carried out through a principal component analysis of these RNAs, observing that exposure to SARS-CoV-2 effectively induces a difference in their expression compared to control cells (Fig 6b). Finally, we investigated whether there was a correlation between the expression of DEGs and lncRNAs with possible cis action. 5 positive correlations were identified between RSAD2 and LINC00487 (r = 0.9989;

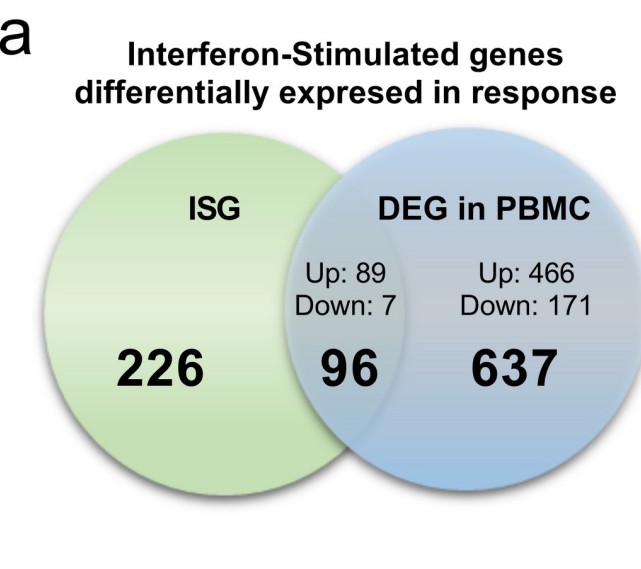

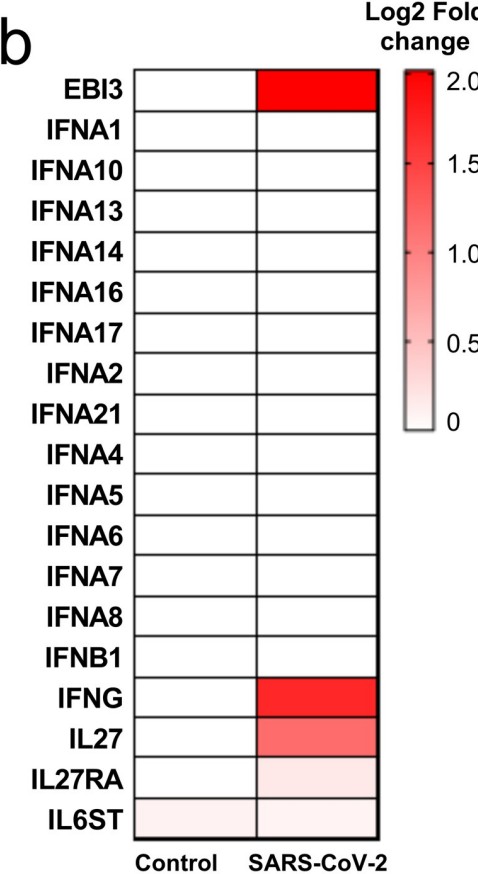

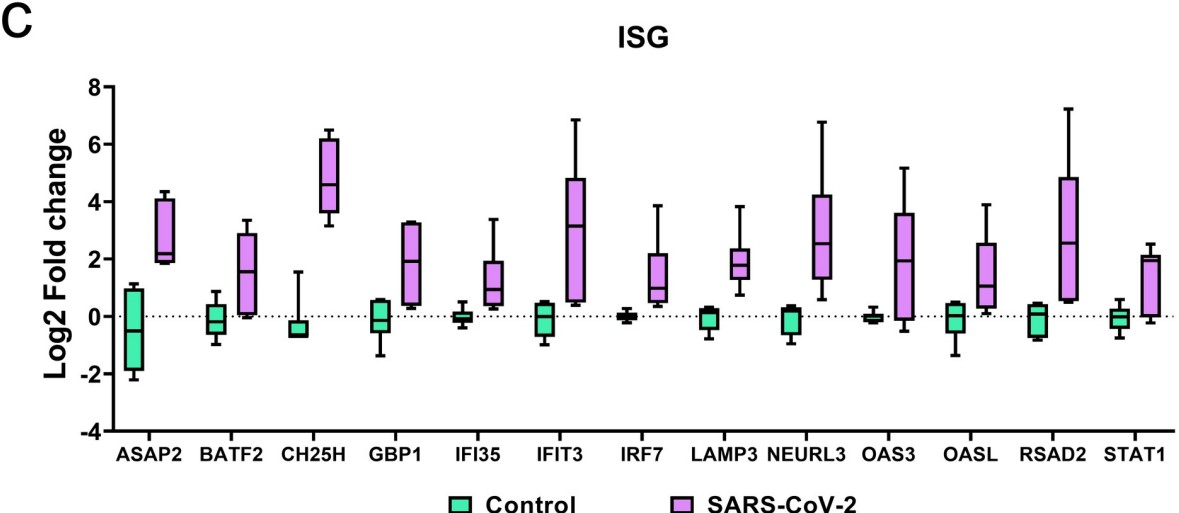

**Fig 3. Exposure to SARS-CoV-2 induces the expression of different interferon response genes.** (a) Differentially expressed interferon response genes to SARS-CoV-2 stimulus within differently expressed genes. (b) Bart plot of interferon response genes represented as log2- fold change (y-axis). (c) Heat map of Z-score differentially expressed interferon genes.

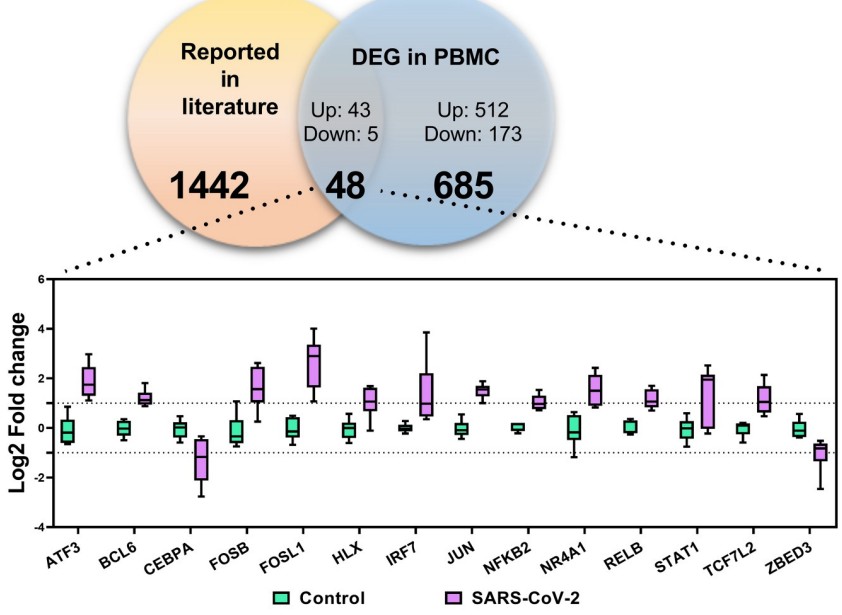

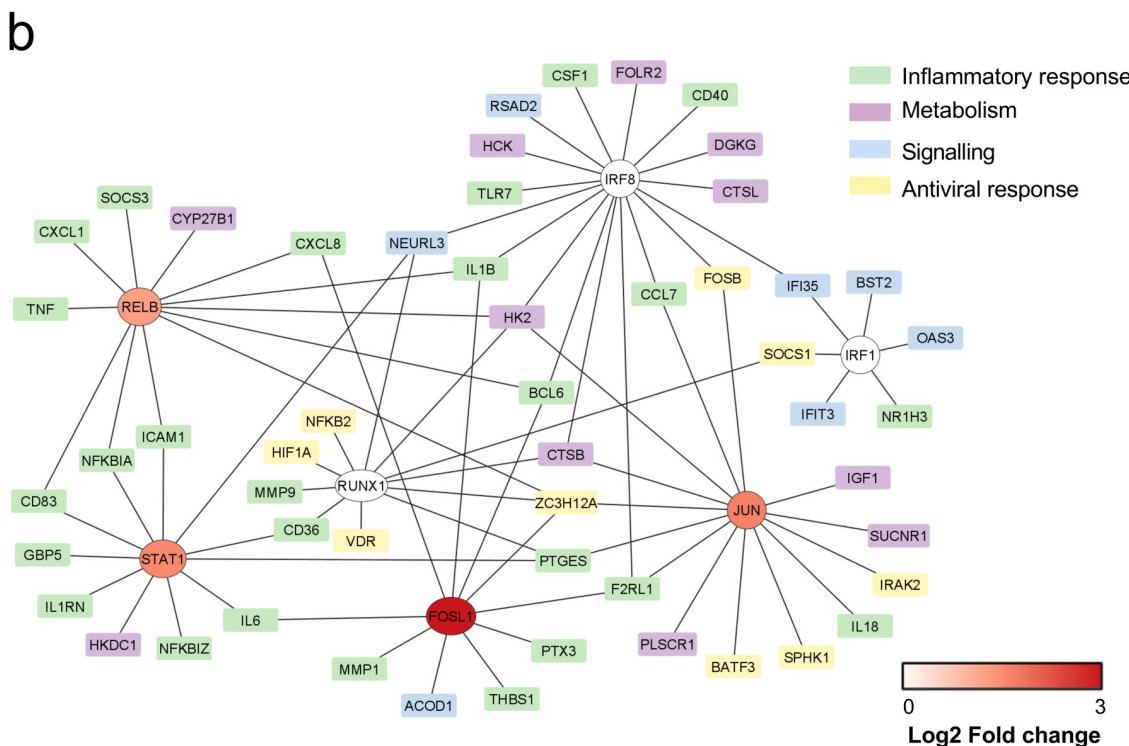

**Fig 4. SARS-CoV-2 regulates the expression of transcriptional factors in PBMCs.** (a) Bar plots of transcriptional factors regulated by SARS-CoV-2 represented as log2- fold change (y-axis). (b) Network of DEG and related transcription factors. DEG are grouped into 4 related processes indicated by a different color. Transcription factors are represented as log2- fold change.

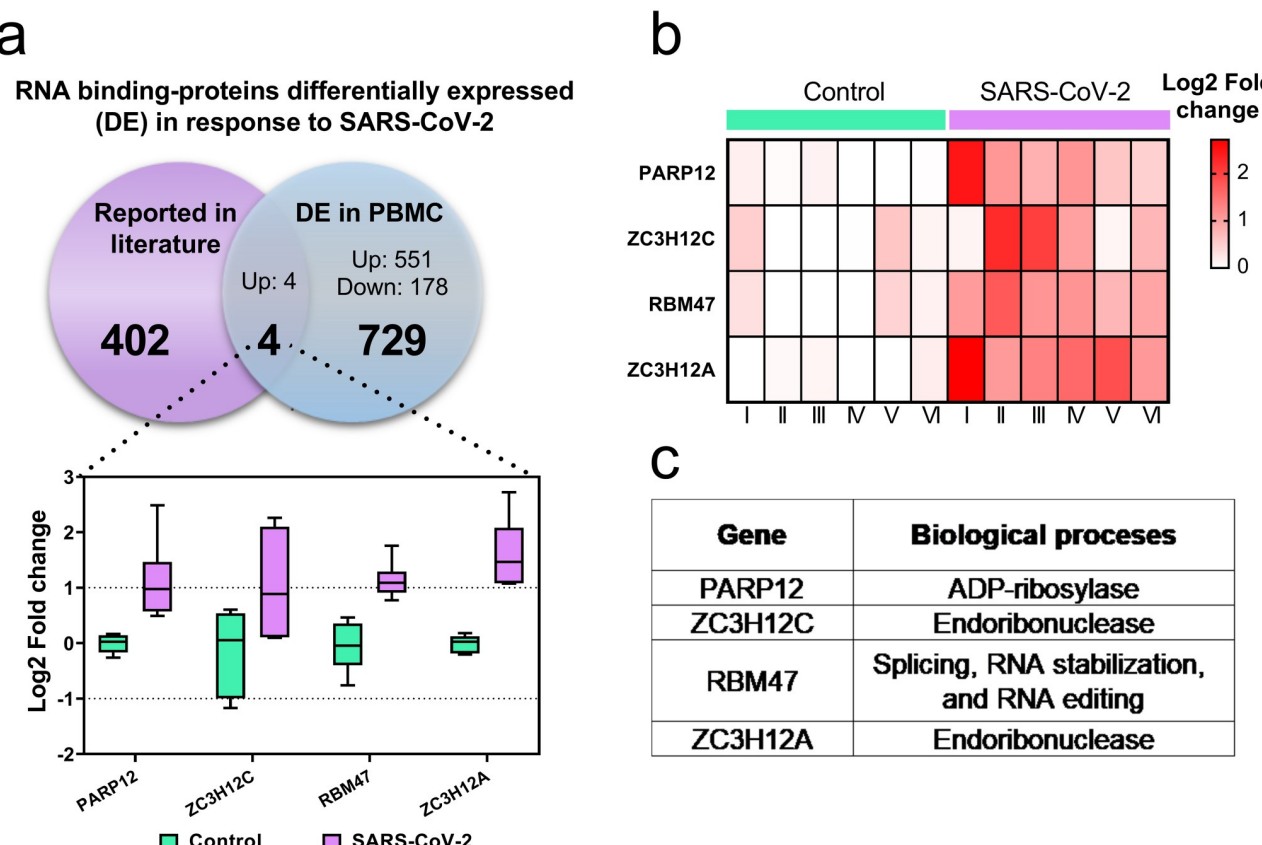

**Fig 5. SARS-CoV-2 regulates the expression of RNA-binding proteins in PBMCs.** (a) Bar plots of transcriptional factors regulated by SARS-CoV-2 represented as log2- fold change (y-axis). (b) Heat map of Z-score normalized differentially expressed RNA binding proteins of control and SARS-CoV-2-exposed cells. (c) Biological function of RNA binding proteins.

Fig 6c), GBP1 and GBP1P1 (r = 0.9282; Fig 6d), GBP4 and GBP1P1 (r = 0.8733; Fig 6e), GBP5 and GBP1P1 (r = 0.9581; Fig 6f), TLR8 and LINC02154 (r = 0.8809; Fig 6g), IL6 and STEAB-P1-AS1 (r = 0.8809; Fig 6h), KLHDC7B and KLHDC7B-DT (0.9808; Fig 6i).

## Discussion

Although the WHO declared the end of the emergency due to the pandemic caused by SARS-CoV-2 and the implementation of vaccination, COVID-19 continues to be a public health problem due to the emergence of new variants and the numerous cases that still occur worldwide [34, 35]. Therefore, it is important to continue investigating the immunopathogenic mechanisms of the virus to identify possible complementary therapeutic targets and biomarkers for early diagnosis or outcomes.

The immunopathogenesis of COVID-19 is complex, where it has been described that the inflammatory response plays an important role in developing severe forms of the disease and fatal outcomes [36], reporting higher levels of inflammatory mediators such as IL-1β, IL-6, and TNF-α in patients requiring ICU admission compared to patients with mild clinical manifestations [10]. Likewise, we previously described that these mediators are produced *in vitro* by PBMCs, indicating that the virus can directly stimulate these cells [37]. Although the infection of PBMCs by SARS-CoV-2 has previously been described [38]; in this study, we found that

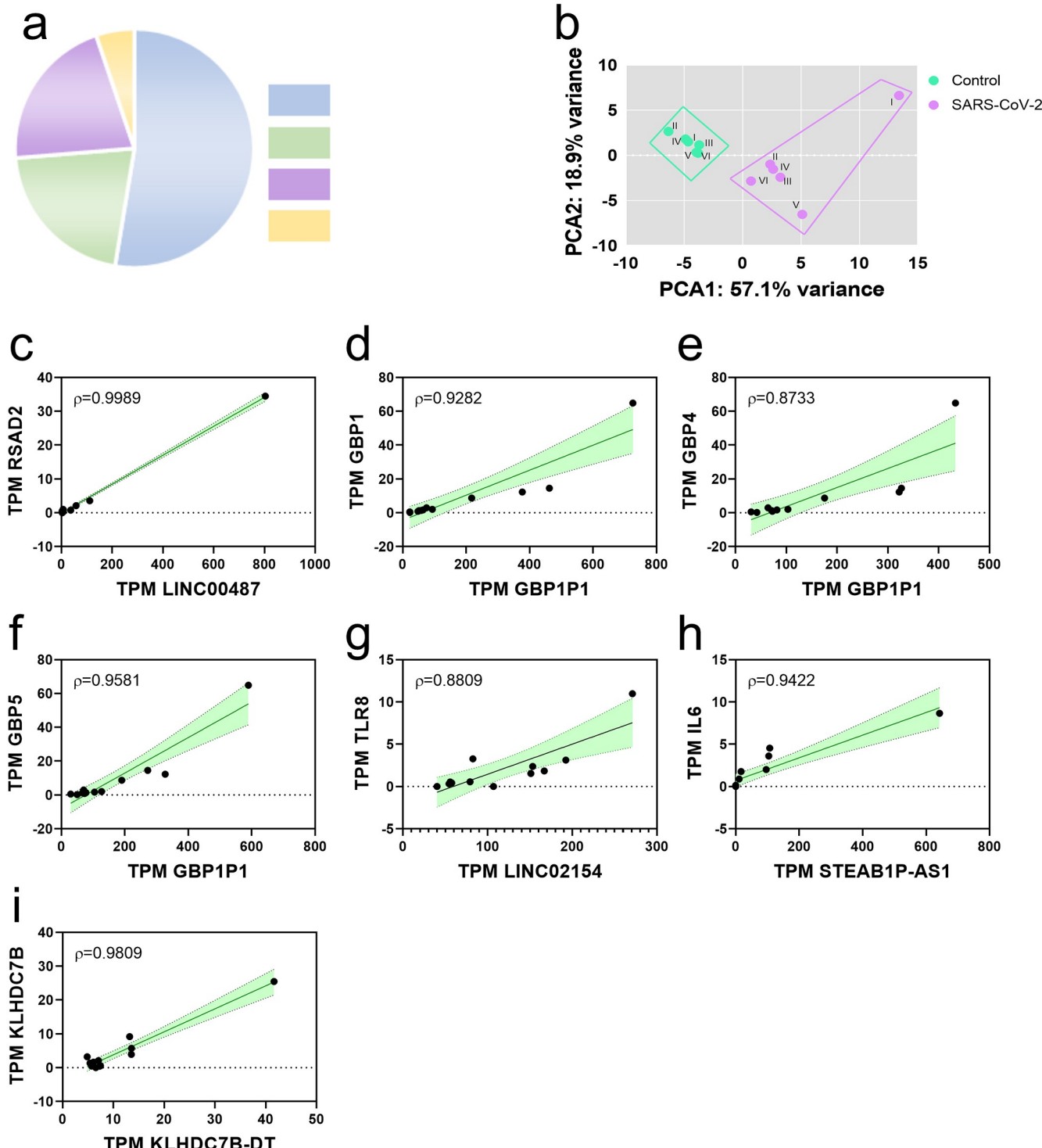

**Fig 6. SARS-CoV-2 regulates the expression of long non-coding RNAs, pseudogenes, antisense RNA and miRNAs.** (a) Total no coding-RNA classes. (b) Principal component analysis of long non-coding RNA expression from PBMCs exposed to SARS-CoV-2. The percentage of the variance of each principal component (PC1 and PC2) for control (I—VI) and SARS-CoV-2 exposed cells (I—VI). (c) Correlations between lcnRNA in cis and white DEG.

SARS-CoV-2 did not have productive replication in PBMCs after 72 h of incubation. This discrepancy may be due to the difference in the experimental procedure. However, RNA-seq analysis revealed that although SARS-CoV-2 does not replicate productively, it can induce a different transcriptional profile than mock cells (control). The regulated genes included genes associated with the inflammatory response, with most being positively regulated. Furthermore, we found that the transcriptional profiles induced by SARS-CoV-2 differ between donors. In particular, the cells of individual 1 presented a greater positive regulation of genes associated with inflammatory and antiviral responses than those of the other donors. In this regard, one study described differences in the transcriptional profile obtained from the whole blood of patients who developed ARDS versus those who did not [39]. This suggests that these differences may contribute to the heterogeneity in the clinical presentation and outcome of patients with COVID-19; thus, studying these genes related to clinical presentation could identify severity predictors and possible intervention targets.

Among genes regulated by SARS-CoV-2, the up regulations of those encoding inflammatory mediators such as IL-6, IL1B, TNF, IL12, CSF1, CSF2, IFNG, CCL2, CCL3, CCL4, CXCL8, CXCL10 was mainly highlighted. Furthermore, an examination of the biological processes associated with the regulated genes revealed that these genes are mainly related to the immune response, with the largest number of associated terms, such as cytokine signaling pathways, the regulation of IL-1β production, neutrophil-mediated immunity, and the mucosal immune response, among others. Additionally, our results indicate that PBMCs contribute to this phenomenon through different mechanisms, including the positive regulation of multiple factors, such as CCL8, ICAM-1, IL-6, IRAK and LLRB2, indicating the migration and activation of different cell populations and highlighting T cells, monocytes, and neutrophils. Among these populations, neutrophils have become relevant for the immune response during viral infection in recent years. In this sense, a significant increase in neutrophils and the formation of extracellular traps (NETs) has been described in patients with severe COVID-19 [40]. An increase in the neutrophil/T cells ratio is observed, an independent predictor of disease severity [40]. Following this, positive feedback of the inflammatory process can be suggested, where PBMCs can produce cytokines and chemokines that can contribute to the continuous recruitment of neutrophils, and these release reactive oxygen species and NETs, exacerbating inflammation.

On the other hand, the regulation of genes involved in amino acid and glucose metabolism must be controlled. In this regard, viruses can alter cellular metabolism, such as through the glucose pathway, which allows them to replicate more efficiently [41]. For example, a study conducted on monocytes infected with SARS-CoV-2 showed greater viral replication and cytokine production in the presence of high levels of glucose [42]. Therefore, we suggest that PBMCs may present an alteration in cellular metabolism that contributes to the outstanding production of inflammatory mediators and, thus, to the pathogenesis of COVID-19.

Next, considering that a specific alteration has been described in establishing the antiviral response, we investigated the expression of interferon and interferon-stimulated genes (ISGs). Induced after SARS-CoV-2 exposure. Specifically, a significant increase in the expression of IFN-γ, IL-27 and EBI3 was observed, suggesting the participation of the IL-27 pathway in the induction of ISGs upon stimulation with SARS-CoV-2 [43–45]. In this regard, it has been described that the IL-27 pathway participates in the response to viral infections through IFN-I-dependent and -independent mechanisms, in addition to triggering the production of different inflammatory mediators [46]. Following this, in this analysis we found positive regulation of genes participating in this pathway, including STAT1, which induces the transcription of ISGs such as OAS3, OASL, IDO1, BTS2, RSAD2, GBP1, GBP4 and GBP5, among others. Additionally, the activation of the IL-27 pathway leads to an increase in the expression

of TLR8, which can recognize viral RNA and activate IRAK kinases, leading to the activation of the transcription factors IRF7 [47], NF-kB and AP-1 and triggering the production of antiviral and inflammatory mediators (Fig 7).

Furthermore, it should be considered that induction of ISGs occurred without a productive viral infection, indicating that bystander cells can enter an antiviral genetic program to resist viral infections. However, a delayed antiviral response has been reported, leading to a high viral load, which, in conjunction with the inflammatory response, can lead to severe forms of the disease [48]. Interestingly, it was observed that cells from individual 1 presented greater

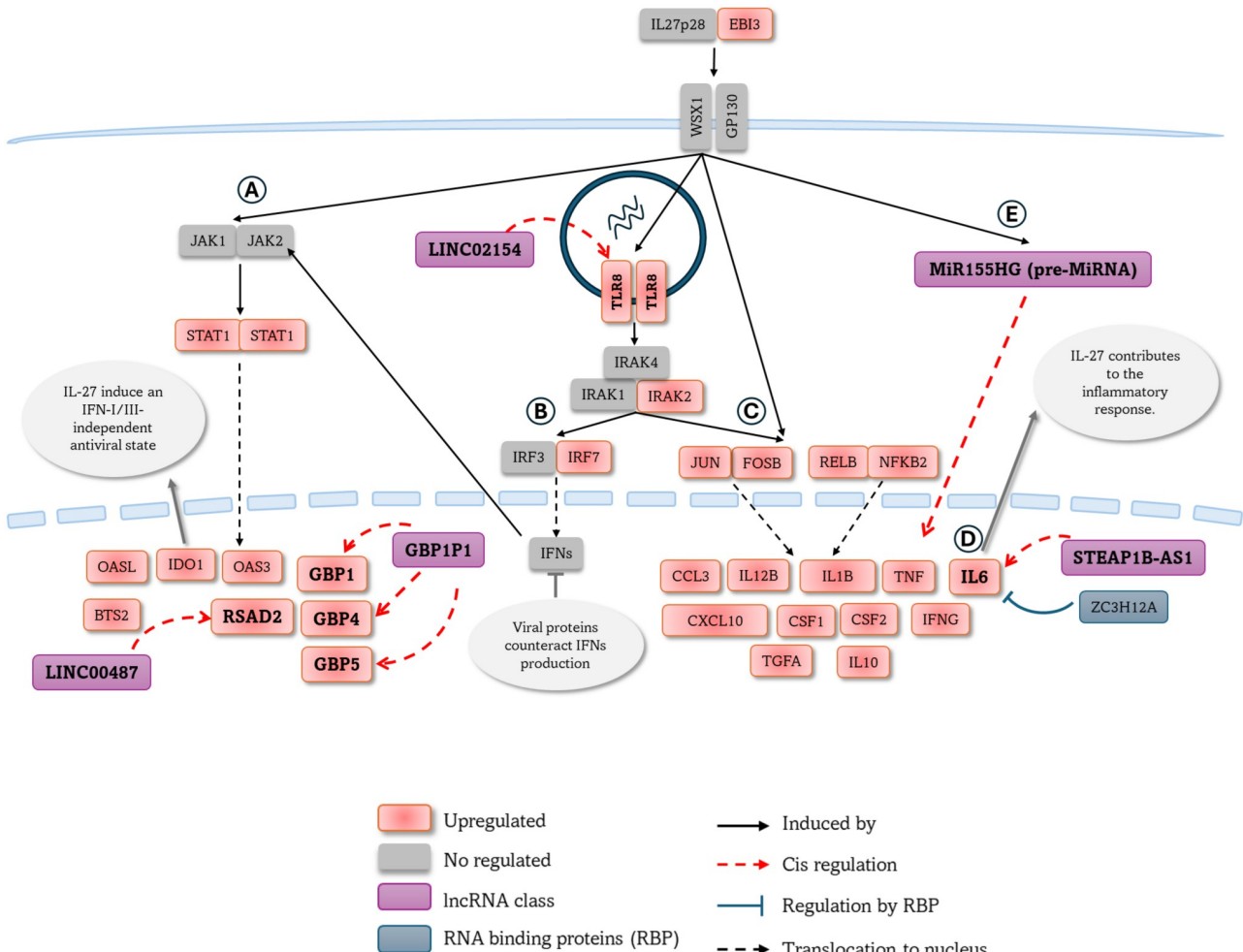

**Fig 7. The lncRNAs induced by SARS-CoV-2 can exert cis-regulation on the IL-27 pathway.** Based on the literature and our RNA-seq analysis, we proposed a schematic representation of the IL-27 pathway during SARS-CoV-2 infection, where IL-27 interacts with its receptor, leading to the activation of signaling pathways. **(A)** IL-27 can directly activate the JAK/STAT signaling pathway, leading to STAT1 homodimerization and induction of different ISGs. In this pathway, lncRNAs LINC00487 and GBP1P1 can upregulate RSAD2 and GBP1/4/5 mRNA, respectively, inducing an IFN-I-independent antiviral state. **(B)** IL-27 promotes TLR8 expression through cis-regulation exerted by LINC02154. TLR8 expression leads to increased detection of viral RNA in endosomes and the activation of kinases such as IRAK4 that can activate IRF7, which in turn is translocated to the nucleus and induces the expression of IFN-I. However, viral proteins can counteract this pathway and prevent or delay the production of these mediators. **(C)** On the other hand, IL-27 may contribute to the inflammatory response through the TLR8/IRAK4 pathway or directly induce the translocation of transcription factors such as AP-1 and NF-kB, leading to the expression of chemokines and cytokines.**(D)** IL-6 has been associated with severe disease and is a target of cis-upregulation by STEAP1B-AS1 and downregulation by the RBP ZC3H12A. **(E)** SARS-CoV-2 induces miRNA expression such as MIR155HG, which can positively regulate the inflammatory response during viral infections. ISG: Inteferon-stimulated gene; miRNA: micro-RNA; lncRNA: Long no coding RNA.

upregulation of these ISGs than other individuals, which led us to consider whether the difference in the expression of these genes may constitute a mechanism that contributes to the clinical heterogeneity of patients with COVID-19; a high expression of ISGs at the beginning of the infection being related to a lower risk of developing severe forms of the disease.

Moreover, multiple signaling pathways are activated during viral infections to trigger an effective response that eliminates the pathogen and restores host homeostasis. Therefore, by investigating the transcription factors (TFs) involved with the described DEGs, we found the regulation of transcription pathways mediated by NF-κB, AP-1, RUNX1, IRF1, IRF8 and STAT1, responsible for inflammation, antiviral response, metabolism and cell signaling. Furthermore, TFs such as NF-kB, AP-1 and STAT-1 can be activated by the IL-27 pathway and trigger an inflammatory and antiviral response. Particularly, NF-κB and FOSL1/JUN (AP-1) promote the transcription of pro-inflammatory cytokines and chemokines, which contribute to the recruitment and activation of inflammatory cell populations [49]. Specifically, it has been proposed that SARS-CoV-2 stimulates the NF-κB activation triggering the production of inflammatory mediators such as TNF-α, IL-1β, MCP-1 [50, 51]. In fact, exacerbated activation of NF-kB has been reported in patients with severe COVID-19 and high levels of IL-6, which highlights the importance of the pathway in the clinical course of the disease [51]. Likewise, it has been described that different components of the AP-1 heterodimer, such as c-JUN and FOSL1, have an important role in various immune system processes [52, 53]. For example, suppression of c-JUN expression efficiently reduced the Influenza virus (IAV) replication, thus restoring the balance between the pro- and anti-inflammatory response [54]. In accordance with these studies and our results, it is proposed that the activation of these TFs leads to the positive regulation of pro-inflammatory cytokines and chemokines such as IL-1β, IL-6, IL-12, TNF-α, CCL7, CCL8, CXCL5, in the PBMCs; which induces greater recruitment of neutrophils, monocytes, and T cells to the sites of infection, thus contributing to an exacerbated inflammatory response and the development of severe symptoms.

Likewise, we observed that parallel to the induction of pro-inflammatory signaling pathways, there was also the activation of anti-inflammatory pathways, a natural mechanism that seeks to prevent the development of exacerbated inflammation. Among these, upregulation of ATF3 was observed, induced during inflammation and genotoxic stress, repressing the expression of target genes by directly binding to their promoters and thus reducing the inflammatory response [55]. For example, ATF3 negatively regulates pro-inflammatory cytokines and chemokines, including IL-6, IL12, MIP-1β [56, 57]x as well as the expression of IFN-γ in NK cells during cytomegalovirus (CMV) infection [58]. Interestingly, ATF3 acts as a negative regulator of antiviral and autophagy cell signaling pathways during Japanese encephalitis virus infection, binding to the promoter region of STAT1, IRF9, ISG15, and ATG5 [59]. In line with this, another identified signaling pathway involves RUNX1, a TF that regulates the differentiation of hematopoietic lineages [60, 61]. One study found that Influenza A (IAV) infection induces the expression of RUNX1 in the alveolar epithelium line A549. Furthermore, suppressing this TF significantly inhibited IAV infection, related to increased production of IFN-β and ISGs. At the same time, overexpression of RUNX1 compromised the expression of these factors and increased the number of virions [62]. These results raise the need to elucidate the role of ATF3 and RUNX1 during SARS-CoV-2 infection, because they could have the potential to modulate the inflammatory response and prevent the development of the cytokine storm. On the other hand, it would be important to consider whether its expression is related to the late antiviral response described during the pathogenesis of COVID-19 and whether it has been related to unfavorable clinical outcomes. Furthermore, it has been suggested that ATF3 is involved in glucose metabolism, an important pathway during viral infections [63, 64]. Therefore,

studying the precise gene response of PBMCs and how this could affect the immune response to SARS-CoV-2 infection would be necessary.

Alternatively, positive regulation of STAT1 was observed, which has a key role in the antiviral response and according to our results is mainly transducing the IL-27 signaling. Activated STAT1 induces the transcription of ISGs, which leads to an antiviral state mediated by the production of proteins that interfere with the viral cycle [64]. For instance, the IFN-γ/STAT1 pathway is involved in the activation of antimicrobial mechanisms like the NADPH oxidase system and inducible nitric oxide synthase (iNOS [65, 66]. Furthermore, among the genes regulated by STAT1 is IL12, which has an interconnection with the IFN signaling pathways and whose main role is stimulating cells to induce Th1 differentiation, which is required for antiviral host defense [67]. Although minimal induction of IFN-I genes was observed, positive regulation of IL27 and IFNG was found, being one of the biological processes to which many DEGs were associated, indicating that this pathway is important during SARS-CoV-2 infection.

On the other hand, our RNA-seq analysis identified the upregulation of RNA-binding proteins and non-coding RNAs, including long non-coding RNAs (lncRNAs), pseudogenes, antisense RNAs, and microRNAs, which regulate gene expression patterns, thus modulating the response to SARS-CoV-2. Among these, lncRNAs (transcripts of more than 200 nucleotides) and antisense RNAs (complementary sequences of a few nucleotides) have important functions in regulating gene expression and have been proposed to play a role during viral infections [68, 69].

Moreover, some of these non-coding RNAs are involved in regulating IL-27 pathway related genes. In this sense, our study found a positive correlation between LINC00487 and the mRNA of RSAD2, an interferon response gene that inhibits RNA virus replication and binding to viral proteins [70]. A significant increase in RSAD2 was previously identified in COVID-19 patients compared to healthy controls. However, in COVID-19 patients requiring the ICU, a significant decrease in RSAD2 was observed compared to those not requiring the ICU, and there was a negative correlation between D-dimer levels and viral load [71]. This indicates that the upregulation of RSAD2 is associated with better clinical outcomes in COVID-19 patients. Furthermore, we observed higher expression of this lncRNA in the cells of individual 1 than in those of the other individuals. This consistent with the differences observed in the expression of ISGs between individuals. Therefore, it is interesting to consider that the cis-regulation of this gene by the long non-coding gene LINC00487 may be related to the response observed in COVID-19 patients and could contribute to the heterogeneity of the clinical presentation in patients; new studies are needed to corroborate this hypothesis.

Similarly, a positive correlation was detected between the GBP1P1 pseudogene and the GBP1, GBP4, and GBP5 mRNAs, indicating that it acts in cis regulating the expression of these ISGs. Guanylate-binding proteins (GBPs) have been described as restriction factors, inhibiting the processing of furin-dependent envelope proteins, affecting virion's infectious capacity [72]. In fact, it has been reported that these ISGs can affect the replication of SARS-CoV-2 [73]. Together with our results, it indicate that the activation of the IL-27 pathway induces the expression of GBP proteins and their positive regulation through non-coding RNAs, which allows establishing an antiviral response that is IFN-I independent.

Activating the IFN pathway induced by TLRs plays a central role in controlling SARS-CoV-2 infection. Among these, TLR8, induced by the IL-27 pathway, can detect double-stranded RNA intermediates during viral replication, leading to the activation of the IRF-7 and NF-κB signaling pathways and triggering the production of IFN-I and pro-inflammatory cytokines, which are necessary to develop an adequate immune response [74]. However, a dual effect of TLRs has been described since their activation can also contribute to the excessive production

of inflammatory mediators [75, 76]. In this analysis, a positive correlation was found between TLR8 mRNA and the lncRNA LINC02154, which can act in cis and regulate the expression of TLR8, indicating that SARS-CoV-2 induces its expression in PBMCs. A study revealed that SARS-CoV-2 can activate neutrophils through TLR8 and produce reactive oxygen species (ROS), cytokines, and NET formation [77]. This, together with regulating chemokines that attract neutrophils, is important for evaluating whether a similar mechanism can occur in monocytes/macrophages and lymphocytes, the main populations of PBMCs.

Regarding inflammatory response, IL-6 has been described as an independent risk predictor for developing severe forms of COVID-19. Interestingly, two mechanisms with opposite effects regulating this cytokine were found in this analysis. On the one hand, a positive correlation of IL-6 mRNA with the antisense RNA STEAP1B-AS1 was observed, indicating a positive cis-regulation. On the other hand, we identified the upregulation of four RNA-binding proteins that can influence gene expression, with ZC3H12A standing out for this role destabilizing IL-6 [78]. An important mechanism in the regulation of gene expression during SARS-CoV-2 infection. Although the role of the exacerbated inflammatory response in developing clinical complications in patients with COVID-19 has been recognized from the beginning, and therapies directed against inflammatory mediators such as IL-1β and IL-6 have been proposed, the results have not yet been conclusive [79, 80]. And these results could suggest that an imbalance in the regulation exerted by these post-transcriptional mechanisms can exacerbate this cytokine's production and favor the development of severe forms of COVID-19. Additionally, the positive regulation of the pre-miRNA MIR155HG was observed, indicating an increase in this miRNA in mature form, which has been reported in different viral infections, promoting the inflammatory response [81, 82]. In addition, it has been described that miRNAs can be induced by the IL-27 pathway during viral infections [83, 84]; therefore, it would be important to investigate whether this miRNA may be being induced by the IL-27 pathway and promoting an inflammatory response.

Finally, KLHDC7B (Kelch domain containing 7B) mRNA, which has no known function but has been described as positively regulated during infectious processes such as human papillomavirus [85], was found to be regulated. Moreover, a study described the positive regulation of this gene during HCV infection. When this gene was silenced using a siRNA, a significant decrease in viral copy number was detected, suggesting that this gene plays an important role in the viral replication of HVC, promoting its replication [86]. In this sense, in our analysis, we found positive regulation of KLHDC7B in PBMCs stimulated with SARS-CoV-2 and even more of a positive correlation with the lncRNA KLHDC7B-DT, suggesting that this lncRNA may promote the expression of KLHDC7B mRNA and thus favor the replication of SARS-CoV-2, which contributes to the development of more severe clinical conditions.

## Limitations

Further studies are required to delineate the specific contributions of each cell subpopulation to the inflammatory process, which could help identify potential therapeutic targets using techniques like single-cell RNA-seq. Additionally, in-depth molecular investigations are needed to explore the role of lncRNAs in the cis-regulation of target RNAs and their potential impact on viral pathogenesis, employing both *in vitro* and *in vivo* models.

## Conclusion

SARS-CoV-2 can modulate the expression of multiple genes in PBMCs even in the absence of productive infection, predominantly affecting pathways related to inflammation and antiviral responses. Regarding the antiviral response, the IL-27 pathway and ISG expression

upregulated, indicating that this pathway may contribute to establishing of the antiviral state during SARS-CoV-2 infection. Furthermore, signaling pathways involved in IL-27 response include the transcription pathways NF-κB, AP-1, and IRFs (Fig 7). This activation produces cytokines and chemokines, which facilitate the recruitment of neutrophils, lymphocytes, and monocytes to the site of infection. Collectively, these findings suggest that PBMCs play a role in the heightened inflammatory response associated with severe clinical outcomes in COVID-19. Additionally, several long non-coding RNAs (lncRNAs) such as LINC00487, LINC02154, GBP1P1, STEAP1B-AS1, and KLHDC7B were identified as regulators of the immune response to SARS-CoV-2 regulating the expression of genes related to the IL-27 pathway. These lncRNAs may contribute to the pathogenesis of severe COVID-19, representing potential therapeutic targets.

## Supporting information

**S1 Fig. Primary genes regulated by SARS-CoV-2 exposure within each term-related to immune response.** Barplot of DEG selected for each term, to show regulated genes in PBMCs exposed to SARS-CoV-2 represented as log2-fold change (y-axis).
(TIF)

**S2 Fig.** (a) Heatmap of Z-score normalized differentially expressed interferon-related genes of control and SARS-CoV-2-exposed cells. (b) Heatmap of Z-score normalized differentially expressed genes (DEG) of control and SARS-CoV-2-exposed cells.
(TIF)

## Author Contributions

**Conceptualization:** Juan C. Hernandez, Natalia Taborda.

**Data curation:** Geysson Javier Fernandez.

**Formal analysis:** Damariz Marin, Geysson Javier Fernandez.

**Funding acquisition:** Juan C. Hernandez, Natalia Taborda.

**Investigation:** Damariz Marin, Geysson Javier Fernandez.

**Methodology:** Damariz Marin, Geysson Javier Fernandez.

**Project administration:** Juan C. Hernandez, Natalia Taborda.

**Resources:** Natalia Taborda.

**Supervision:** Juan C. Hernandez.

**Writing – original draft:** Damariz Marin, Juan C. Hernandez.

**Writing – review & editing:** Juan C. Hernandez, Natalia Taborda.

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
