## [Decision Letter · Decision Letter 0]

30 Sep 2024

PONE-D-24-35955A systems biology approach unveils different gene expression control mechanisms governing the immune response genetic program in peripheral blood mononuclear cells exposed to SARS-CoV-2PLOS ONE

Dear Dr. Hernandez,

Thank you for submitting your manuscript to PLOS ONE. After careful consideration, we feel that it has merit but does not fully meet PLOS ONE’s publication criteria as it currently stands. Therefore, we invite you to submit a revised version of the manuscript that addresses the points raised during the review process.

 Editor's comments: 1. Line 50-53: "This complex respiratory disease presents a myriad of potentially life-threatening complications, including pneumonia, acute respiratory distress syndrome (ARDS), acute cardiac injury, disseminated intravascular coagulation, and multiorgan failure [1]." More references are needed, with this one (PMID: 34406882) as examples (citing is optional). 2. Line 60-61: "In light of these challenges, it is important to consider the pathogenesis of COVID-19, particularly its association with inflammatory processes". There are no references to support this statement. More references are needed, with these two (PMID: 33667962 and 33337932) as examples (citing is optional). 3. What are the inclusion and exclusion criteria of enrolled cases? 4. Line 486-487: "a dual effect of TLRs has been described since their activation can also contribute to the excessive production of inflammatory mediators": There are no references to support this statement. More references are needed, with these two (PMID: 38556084 and 22948160) as examples (citing is optional).

We look forward to receiving your revised manuscript.

Kind regards,

Benjamin M. Liu, MBBS, PhD, D(ABMM), MB(ASCP)

Academic Editor

PLOS ONE

Journal requirements: 1. When submitting your revision, we need you to address these additional requirements.Please ensure that your manuscript meets PLOS ONE's style requirements, including those for file naming. The PLOS ONE style templates can be found at https://journals.plos.org/plosone/s/file?id=wjVg/PLOSOne_formatting_sample_main_body.pdf and https://journals.plos.org/plosone/s/file?id=ba62/PLOSOne_formatting_sample_title_authors_affiliations.pdf 2. Please include a caption for figure 3a. 3. We noticed you have some minor occurrence of overlapping text with the following previous publication(s), which needs to be addressed: https://doi.org/10.1038/s41598-023-39921-w In your revision ensure you cite all your sources (including your own works), and quote or rephrase any duplicated text outside the methods section. Further consideration is dependent on these concerns being addressed. 4. Thank you for stating the following financial disclosure:  [This work was supported by Universidad Cooperativa de Colombia, Universidad de Antioquia and Corporación Universitaria Remington.].  Please state what role the funders took in the study.  If the funders had no role, please state: ""The funders had no role in study design, data collection and analysis, decision to publish, or preparation of the manuscript."" If this statement is not correct you must amend it as needed. Please include this amended Role of Funder statement in your cover letter; we will change the online submission form on your behalf. 5. Thank you for stating the following in the Acknowledgments Section of your manuscript: [The authors would like to thank all health participants and to Dr Jorge Humberto Tabares for his technical support. Universidad de Antioquia UdeA, Universidad Cooperativa de Colombia and Corporación universitaria Remington. ]We note that you have provided funding information that is not currently declared in your Funding Statement. However, funding information should not appear in the Acknowledgments section or other areas of your manuscript. We will only publish funding information present in the Funding Statement section of the online submission form. Please remove any funding-related text from the manuscript and let us know how you would like to update your Funding Statement. Currently, your Funding Statement reads as follows:  [This work was supported by Universidad Cooperativa de Colombia, Universidad de Antioquia and Corporación Universitaria Remington.].  Please include your amended statements within your cover letter; we will change the online submission form on your behalf. 6. Please include captions for your Supporting Information files at the end of your manuscript, and update any in-text citations to match accordingly. Please see our Supporting Information guidelines for more information: http://journals.plos.org/plosone/s/supporting-information. 

Reviewers' comments:

Reviewer's Responses to Questions

**Comments to the Author**

1. Is the manuscript technically sound, and do the data support the conclusions?

Reviewer #1: No

Reviewer #2: Yes

2. Has the statistical analysis been performed appropriately and rigorously? 

Reviewer #1: No

Reviewer #2: Yes

3. Have the authors made all data underlying the findings in their manuscript fully available?

Reviewer #1: No

Reviewer #2: Yes

4. Is the manuscript presented in an intelligible fashion and written in standard English?

Reviewer #1: No

Reviewer #2: Yes

5. Review Comments to the Author

Reviewer #1: The authors used RNA- seq analysis to profile human peripheral blood mononuclear cells (PBMCs) from healthy individuals after in vitro stimulation with SARS-CoV-2 compared to unexposed cells. The authors highlighted that SARS-CoV-2 induced the expression of chemokines involved in the recruitment of T-cells, neutrophils and monocytes. Also, the authors emphasized that the transcription factors associated with inflammatory pathways (e.g., JUN, RELB, NFKB2, etc.) and lncRNA, involved in cis-regulation of different genes were differentially expressed. Finally the authors suggested that these pathways might be possible therapeutic targets.

Overall, the manuscript addresses an important aspect of SARS-CoV-2 infection and its impact on the immune system. However, the contribution of the study’s results to current knowledge is not clearly articulated.The authors need to revise the introduction to clearly state the knowledge gap that prompted this study, specifying whether it pertains to specific changes in the virus, a particular population, or other factors.

Similar studies have been published since start of COVID-19 pandemic using comparable techniques and analyzing responses in PBMCs and other human cells, often with larger sample sizes.Therefore, I recommend that the authors include the study's aim in the abstract, revise the introduction to specify the study’s objective and the gap in knowledge it addresses, and update the discussion to highlight how their results contribute to existing research.

Both the introduction and discussion sections should include a broader range of studies, as only a few relevant works are currently cited. The authors need to emphasize how their findings advance our understanding of SARS-CoV-2’s effects on the human immune response. I included at the end of this review a list of a few studies as examples from various groups all over the world.

Additionally, the abstract is unclear regarding whether the PBMCs were from healthy individuals previously exposed to SARS-CoV-2 or PBMCs were exposed to the virus in vitro during the study. The methods section mentions that the PBMCs were obtained from six healthy male donors, so this needs to be clarified in the abstract.

The methodology is not sufficiently detailed for replication, particularly in the statistical analyses and the core data analysis involving differential gene expression and transcription factor motif analysis. The programs used were not specified, nor was it mentioned whether R Studio or any relevant code and tools were used.

Furthermore, the authors Figshare link provided is broken. The authors must provide a working link for the benefit of reviewers.

The study’s results do not provide mechanistic insights, as they are solely based on RNA-seq data. Additional mechanistic experiments are needed.

Finally, Figure 4B has poor-quality graphics. A higher-resolution version should be provided.

Below are suggested studies that should be included in the results and discussion sections of the manuscript:

Zhu, L., Yang, P., Zhao, Y., Zhuang, Z., Wang, Z., Song, R., Zhang, J., Liu, C., Gao, Q., Xu, Q., Wei, X., Sun, H. X., Ye, B., Wu, Y., Zhang, N., Lei, G., Yu, L., Yan, J., Diao, G., Meng, F., … Liu, W. J. (2020). Single-Cell Sequencing of Peripheral Mononuclear Cells Reveals Distinct Immune Response Landscapes of COVID-19 and Influenza Patients. Immunity, 53(3), 685–696.e3. https://doi.org/10.1016/j.immuni.2020.07.009

Stephenson, E., Reynolds, G., Botting, R.A. et al. Single-cell multi-omics analysis of the immune response in COVID-19. Nat Med 27, 904–916 (2021). https://doi.org/10.1038/s41591-021-01329-2

Fischer, D.S., Ansari, M., Wagner, K.I. et al. Single-cell RNA sequencing reveals ex vivo signatures of SARS-CoV-2-reactive T cells through ‘reverse phenotyping’. Nat Commun 12, 4515 (2021). https://doi.org/10.1038/s41467-021-24730-4

Kashima, Y., Mizutani, T., Nakayama-Hosoya, K., Moriyama, S., Matsumura, T., Yoshimura, Y., Sasaki, H., Horiuchi, H., Miyata, N., Miyazaki, K., Tachikawa, N., Takahashi, Y., Suzuki, T., Sugano, S., Matano, T., Kawana-Tachikawa, A., & Suzuki, Y. (2023). Multimodal single-cell analyses of peripheral blood mononuclear cells of COVID-19 patients in Japan. Scientific reports, 13(1), 1935. https://doi.org/10.1038/s41598-023-28696-9

Reviewer #2: Authors presented the gene expression data on the exposure of PBMCs from healthy individuals to SARS-CoV2 virus. Following are the observations on the MS:

1. Number of individuals used in this study is low.

2. What is the new data obtained in this study when compared to similar studies already conducted?

3. Lack of laboratory proofs for the gene expression analysis apart from real-time PCR data.

6. PLOS authors have the option to publish the peer review history of their article (what does this mean?). If published, this will include your full peer review and any attached files.

Reviewer #1: No

Reviewer #2: No

---

## [Author Response · Author response to Decision Letter 0]

14 Nov 2024

Dear Dr. Editor,

Enclosed is the revised version of our manuscript PONE-D-24-35955, entitled “A systems biology approach unveils different gene expression control mechanisms governing the immune response genetic program in peripheral blood mononuclear cells exposed to SARS-CoV-2”. Each point raised by the reviewer was considered and carefully reviewed. Changes were included in the manuscript, as described below. 

Editor's comments:

1. Line 50-53: "This complex respiratory disease presents a myriad of potentially life-threatening complications, including pneumonia, acute respiratory distress syndrome (ARDS), acute cardiac injury, disseminated intravascular coagulation, and multiorgan failure [1]." More references are needed, with this one (PMID: 34406882) as examples (citing is optional).

AUTHOR RESPONSE AND ACTION TAKEN: 

The new references were added as suggested.

2. Line 60-61: "In light of these challenges, it is important to consider the pathogenesis of COVID-19, particularly its association with inflammatory processes". There are no references to support this statement. More references are needed, with these two (PMID: 33667962 and 33337932) as examples (citing is optional).

AUTHOR RESPONSE AND ACTION TAKEN: 

The new references were added as suggested.

3. What are the inclusion and exclusion criteria of enrolled cases?

AUTHOR RESPONSE AND ACTION TAKEN: 

The criteria was included in the text as follows:

“Six healthy male volunteers between 20 and 40 years old were included. Individuals who reported any illness in the previous 4 weeks, chronic diseases, long-term medication use, cancer, illicit drug use or mental disorders were excluded”.

4. Line 486-487: "a dual effect of TLRs has been described since their activation can also contribute to the excessive production of inflammatory mediators": There are no references to support this statement. More references are needed, with these two (PMID: 38556084 and 22948160) as examples (citing is optional).

AUTHOR RESPONSE AND ACTION TAKEN: 

The new references were added as suggested.

AUTHOR RESPONSE AND ACTION TAKEN: 

The manuscript was double-checked to meet the PLOS ONE's style requirements.

2. Please include a caption for figure 3a.

AUTHOR RESPONSE AND ACTION TAKEN: 

The caption was included in figure 3a: “Interferon-Stimulated genes differentially expressed (DE) in response to SARS-CoV-2”

3. We noticed you have some minor occurrence of overlapping text with the following previous publication(s), which needs to be addressed: https://doi.org/10.1038/s41598-023-39921-w

 In your revision ensure you cite all your sources (including your own works), and quote or rephrase any duplicated text outside the methods section. Further consideration is dependent on these concerns being addressed.

AUTHOR RESPONSE AND ACTION TAKEN: 

All the sources were cited in the manuscript, and some parts were rephrased to avoid duplicated text. In addition, Turnitin was used to evaluate the similarity (report attached). 

 [This work was supported by Universidad Cooperativa de Colombia, Universidad de Antioquia and Corporación Universitaria Remington.]. 

AUTHOR RESPONSE AND ACTION TAKEN: 

The change has been made as indicated. 

AUTHOR RESPONSE AND ACTION TAKEN: 

It was addressed.

[The authors would like to thank all health participants and to Dr Jorge Humberto Tabares for his technical support. Universidad de Antioquia UdeA, Universidad Cooperativa de Colombia and Corporación universitaria Remington. ]

 [This work was supported by Universidad Cooperativa de Colombia, Universidad de Antioquia and Corporación Universitaria Remington.]. 

AUTHOR RESPONSE AND ACTION TAKEN: 

It was addressed.

AUTHOR RESPONSE AND ACTION TAKEN: 

The change has been made and the Funding information has been removed from the Acknowledgments section. 

AUTHOR RESPONSE AND ACTION TAKEN: 

The captions were included at the end of the manuscript

Reviewers' comments:

Reviewer #1: 

The authors used RNA- seq analysis to profile human peripheral blood mononuclear cells (PBMCs) from healthy individuals after in vitro stimulation with SARS-CoV-2 compared to unexposed cells. The authors highlighted that SARS-CoV-2 induced the expression of chemokines involved in the recruitment of T-cells, neutrophils and monocytes. Also, the authors emphasized that the transcription factors associated with inflammatory pathways (e.g., JUN, RELB, NFKB2, etc.) and lncRNA, involved in cis-regulation of different genes were differentially expressed. Finally the authors suggested that these pathways might be possible therapeutic targets.

1. Overall, the manuscript addresses an important aspect of SARS-CoV-2 infection and its impact on the immune system. However, the contribution of the study’s results to current knowledge is not clearly articulated. The authors need to revise the introduction to clearly state the knowledge gap that prompted this study, specifying whether it pertains to specific changes in the virus, a particular population, or other factors.

Similar studies have been published since start of COVID-19 pandemic using comparable techniques and analyzing responses in PBMCs and other human cells, often with larger sample sizes. Therefore, I recommend that the authors include the study's aim in the abstract, revise the introduction to specify the study’s objective and the gap in knowledge it addresses, and update the discussion to highlight how their results contribute to existing research.

AUTHOR RESPONSE AND ACTION TAKEN: 

The following aim was included in the abstract: 

“The RNA-seq technique to analyze mRNA and non-coding RNA profiles of human peripheral blood mononuclear cells (PBMCs) from healthy individuals after SARS-CoV-2 in vitro exposure, to identify pathways related to immune response and the regulatory posttranscriptional mechanisms triggered that can serve as possible complementary therapeutic targets.”

The Introduction and Discussion sections were updated as indicated.

2. Both the introduction and discussion sections should include a broader range of studies, as only a few relevant works are currently cited. The authors need to emphasize how their findings advance our understanding of SARS-CoV-2’s effects on the human immune response. I included at the end of this review a list of a few studies as examples from various groups all over the world.

AUTHOR RESPONSE AND ACTION TAKEN: 

The manuscript includes new references and other studies that complement the changes made so far.

3. Additionally, the abstract is unclear regarding whether the PBMCs were from healthy individuals previously exposed to SARS-CoV-2 or PBMCs were exposed to the virus in vitro during the study. The methods section mentions that the PBMCs were obtained from six healthy male donors, so this needs to be clarified in the abstract.

AUTHOR RESPONSE AND ACTION TAKEN: 

The following was included:

“…RNA-seq technique to analyze mRNA and non-coding RNA profiles of human peripheral blood mononuclear cells (PBMCs) from healthy individuals after SARS-CoV-2 in vitro exposure”

The methodology is not sufficiently detailed for replication, particularly in the statistical analyses and the core data analysis involving differential gene expression and transcription factor motif analysis. The programs used were not specified, nor was it mentioned whether R Studio or any relevant code and tools were used.

AUTHOR RESPONSE AND ACTION TAKEN: 

The next information was included:

Briefly, data were preprocessed with Trimmomatic and Prinseq functions, setting an average Phred score >20. Subsequently, alignment with the reference genome (Homosapiens GRch38.14) was performed with the TopHat2 function and counts for RefSeq genes were obtained with HTseq. The content matrix was normalized using the DESeq2 v.3.20 package in R v.4.2.2.

Furthermore, the authors Figshare link provided is broken. The authors must provide a working link for the benefit of reviewers.

AUTHOR RESPONSE AND ACTION TAKEN: 

The link has an embargo while the manuscript is published. However, we have attached the dataset, only for reviewers.

The study’s results do not provide mechanistic insights, as they are solely based on RNA-seq data. Additional mechanistic experiments are needed.

AUTHOR RESPONSE AND ACTION TAKEN: 

We agree with reviewer. However, the grant has expired, and no additional funds are available to conduct new experiments. As stated in the Limitations section, additional studies are necessary to understand the role of the regulating mechanisms during the COVID-19 pathogenesis and their potential for future therapeutic approaches.

Finally, Figure 4B has poor-quality graphics. A higher-resolution version should be provided.

AUTHOR RESPONSE AND ACTION TAKEN: 

The figure was provided in TIF with a resolution of 600 dpi

Below are suggested studies that should be included in the results and discussion sections of the manuscript:

Zhu, L., Yang, P., Zhao, Y., Zhuang, Z., Wang, Z., Song, R., Zhang, J., Liu, C., Gao, Q., Xu, Q., Wei, X., Sun, H. X., Ye, B., Wu, Y., Zhang, N., Lei, G., Yu, L., Yan, J., Diao, G., Meng, F., … Liu, W. J. (2020). Single-Cell Sequencing of Peripheral Mononuclear Cells Reveals Distinct Immune Response Landscapes of COVID-19 and Influenza Patients. Immunity, 53(3), 685–696.e3. https://doi.org/10.1016/j.immuni.2020.07.009

Stephenson, E., Reynolds, G., Botting, R.A. et al. Single-cell multi-omics analysis of the immune response in COVID-19. Nat Med 27, 904–916 (2021). https://doi.org/10.1038/s41591-021-01329-2

Fischer, D.S., Ansari, M., Wagner, K.I. et al. Single-cell RNA sequencing reveals ex vivo signatures of SARS-CoV-2-reactive T cells through ‘reverse phenotyping’. Nat Commun 12, 4515 (2021). https://doi.org/10.1038/s41467-021-24730-4

Kashima, Y., Mizutani, T., Nakayama-Hosoya, K., Moriyama, S., Matsumura, T., Yoshimura, Y., Sasaki, H., Horiuchi, H., Miyata, N., Miyazaki, K., Tachikawa, N., Takahashi, Y., Suzuki, T., Sugano, S., Matano, T., Kawana-Tachikawa, A., & Suzuki, Y. (2023). Multimodal single-cell analyses of peripheral blood mononuclear cells of COVID-19 patients in Japan. Scientific reports, 13(1), 1935. https://doi.org/10.1038/s41598-023-28696-9

AUTHOR RESPONSE AND ACTION TAKEN: 

The new references were added as suggested.

Reviewer #2: Authors presented the gene expression data on the exposure of PBMCs from healthy individuals to SARS-CoV2 virus. Following are the observations on the MS:

1. Number of individuals used in this study is low.

AUTHOR RESPONSE AND ACTION TAKEN: 

Although the number of individuals is not high, taking into account the reading depth of the sequencing, the statistical tests applied to the data and the parameters to consider statistically significant differences (FDR <0.05 and a Foldchange of 2) it can be assumed that the results obtained are reliable. Furthermore, when reviewing the scientific literature, we found that our results are consistent with what has been previously published. Furthermore, other studies have used a similar or even lower number of individuals, obtaining relevant results.

Goswami S, Hu X, Chen Q, Qiu J, Yang J, Poudyal D, Sherman BT, Chang W, Imamichi T. Profiles of MicroRNAs in Interleukin-27-Induced HIV-Resistant T Cells: Identification of a Novel Antiviral MicroRNA. J Acquir Immune Defic Syndr. 2021 Mar 1;86(3):378-387. doi: 10.1097/QAI.0000000000002565. PMID: 33196551; PMCID: PMC7879852.

Fernandez GJ, Ramirez-Mejia JM, Urcuqui-Inchima S. Transcriptional and post-transcriptional mechanisms that regulate the genetic program in Zika virus-infected macrophages. Int J Biochem Cell Biol. 2022;153:106312.

2. What is the new data obtained in this study when compared to similar studies already conducted?

AUTHOR RESPONSE AND ACTION TAKEN: 

While previous studies have described the transcriptional profile of PBMCs exposed in vitro to SARS-CoV-2 and have reported the upregulation of multiple inflammatory and antiviral genes, in our study non-coding RNAs, such as long-non-coding RNAs and antisense RNAs, were identified, which constitute mechanisms for regulating gene expression. Furthermore, when exploring the genes that the cis-regulation of these non-coding RNAs may target, we found the IL-27 pathway. This pathway triggers the production of antiviral mediators, which may be independent of IFN-I and III, constituting an important alternative pathway to overcome the immune evasion mechanisms developed by the virus. In addition, we observed that the response between donors is heterogeneous, including the expression of these non-coding RNAs, which could contribute to the differences in clinical presentation observed between patients. Therefore, these lnc-RNA and genes could constitute a complementary therapeutic target.

3. Lack of laboratory proofs for the gene expression analysis apart from real-time PCR data.

AUTHOR RESPONSE AND ACTION TAKEN: 

We agree with reviewer. However, the grant has expired, and no additional funds are available to conduct new experiments. As stated in the Limitations section, additional studies are necessary to understand the role of the regulating mechanisms during the COVID-19 pathogenesis and their potential for future therapeutic approaches.

Finally, we want to thank the reviewers for their pertinent and useful comments, and the Editor for the opportunity to resubmit our work. We are convinced that all the changes have significantly improved the manuscript.

---

## [Editor Report · Decision Letter 1]

18 Nov 2024

A systems biology approach unveils different gene expression control mechanisms governing the immune response genetic program in peripheral blood mononuclear cells exposed to SARS-CoV-2

PONE-D-24-35955R1

Dear Dr. Hernandez,

We’re pleased to inform you that your manuscript has been judged scientifically suitable for publication and will be formally accepted for publication once it meets all outstanding technical requirements.

Kind regards,

Benjamin M. Liu, MBBS, PhD, D(ABMM), MB(ASCP)

Academic Editor

PLOS ONE
---

## [Editor Report · Acceptance letter]

26 Nov 2024

PONE-D-24-35955R1 

PLOS ONE

Dear Dr. Hernandez, 

I'm pleased to inform you that your manuscript has been deemed suitable for publication in PLOS ONE. Congratulations! Your manuscript is now being handed over to our production team.

Kind regards, 

on behalf of

Dr. Benjamin M. Liu 

Academic Editor

PLOS ONE